# Quantification of Protein Uptake by Endocytosis in Carnivorous Nepenthales

**DOI:** 10.3390/plants12020341

**Published:** 2023-01-11

**Authors:** Caroline Ivesic, Stefanie Krammer, Marianne Koller-Peroutka, Aicha Laarouchi, Daniela Gruber, Ingeborg Lang, Irene K. Lichtscheidl, Wolfram Adlassnig

**Affiliations:** 1Functional and Evolutionary Ecology, Faculty of Life Sciences, University of Vienna, Djerassiplatz 1, 1030 Vienna, Austria; 2Core Facility Cell Imaging and Ultrastructure Research, Faculty of Life Sciences, University of Vienna, Djerassiplatz 1, 1030 Vienna, Austria

**Keywords:** *Drosophyllum*, *Dionaea*, *Drosera roseana*, *Drosera capensis*, *Nepenthes*, endosome, protein uptake, FITC-BSA, 3D modeling of glands

## Abstract

Carnivorous plants adsorb prey-derived nutrients partly by endocytosis. This study quantifies endocytosis in *Drosophyllum lusitanicum*, *Drosera capensis*, *Drosera roseana*, *Dionaea muscipula* and *Nepenthes* × *ventrata*. Traps were exposed to 1% fluorescent-labeled albumin (FITC-BSA), and uptake was quantified repeatedly for 64 h. Formation of vesicles started after ≤1 h in adhesive traps, but only after 16 h in species with temporary stomach (*D. muscipula* and *N*. × *ventrata*). In general, there are similarities in the observed species, especially in the beginning stages of endocytosis. Nonetheless, further intracellular processing of endocytotic vesicles seems to be widely different between species. Endocytotic vesicle size increased significantly over time in all species except in *D. capensis*. Fluorescence intensity of the endocytotic vesicles increased in all species except *D. muscipula*. After 64 h, estimates for FITC-BSA absorption per gland ranged from 5.9 ± 6.3 ng in *D. roseana* to 47.8 ± 44.3 ng in *N.* × *ventrata*, demonstrating that endocytosis substantially contributes to the adsorption of prey-derived nutrients.

## 1. Introduction

Carnivorous plants (CPs) attract, trap, degrade, and absorb tiny animals in order to supplement their nutrient supply [1,2]. Prey digestion and uptake in CPs is functionally similar to the processes observed in the digestive systems of animals [3,4]. However, in CPs the functional features are all performed in one spot: the trap leaf. Digestion and uptake are accomplished either by enzymes produced by the plant itself, or by mutualistic trap inquilines [4,5]. In both cases, prey degradation results in a complex mixture of inorganic ions such as K^+^ or PO_4_^3−^ [6], small organic molecules such as amino-sugars from chitin, amino acids, peptides of various size, and macromolecules including whole proteins [7,8,9]. Regardless of their composition, uptake of prey-derived nutrients always follows the same principal pathway [10]: Uptake is performed by specialized cells of the traps, which form morphologically differentiated glands in the majority of CPs [2,11]. Nutrients enter the apoplast of these cells via the very same pores in the otherwise impermeable cuticle of the trap, which are also used for the secretion of digestive enzymes, pitcher fluid or adhesive mucilage [11]. Subsequently, transport via the cytoplasm is enforced by an endodermis that seals the apoplast of the gland from the apoplast of the interior of the leaf [2]. In order to facilitate this transfer of nutrients from the apoplast into the living cytoplasm, glandular cells are usually equipped by an increased contact surface between the cell wall and the plasma membrane [12,13].

Uptake from the apoplast into the cytoplasm is usually believed to be performed by selective, active or passive membrane carriers [3,14,15]. Such carriers would be very efficient in the uptake of ions and small molecules, possibly up to the size of oligopeptides as in *Sarracenia* [16]. Uptake by carriers, however, would require a very thorough digestion of the prey animals, since such carriers are hardly able to handle a multitude of merely degraded macromolecules. Therefore, it was early hypothesized [17,18] that carriers might be supplemented by endocytosis, which would enable the uptake of molecules of virtually any kind and size. In an earlier publication, the authors of this study finally demonstrated endocytotic nutrient uptake in Nepenthaceae, Droseraceae, Drosophyllaceae, but also in some Lentibulariaceae, Cephalotaceae and Sarraceniaceae [19,20,21]. Thus, endocytosis seems to be ubiquitous in those highly evolved CPs, which are usually placed in the order Nepenthales. Here, uptake of endocytotic markers revealed a common pattern, which differed clearly from the situation of unrelated species such as *Genlisea violacea* × *lobata* (Lentibulariaceae), *Sarracenia oreophila* (Sarraceniaceae) or *Cephalotus follicularis* (Cephalotaceae): After stimulation, a multitude of tiny endocytotic vesicles is formed, which typically increase in size but decrease in number over time. This suggests fusion with lysosomes to form large endosomes, where intracellular digestion of the absorbed nutrients seems to take place.

Nepenthales are a taxonomically and morphologically diverse order, containing four families with a total of approximately 400 species, using passive adhesive traps (*Triphyophyllum* and *Drosophyllum*), active adhesive traps (*Drosera*), pitcher traps (*Nepenthes*) and highly mobile snap traps (*Dionaea* and *Aldrovanda*) [22]. It is yet unknown if and how uptake mechanisms differ between systematic groups and trap types. This study compares a representative sample of species from Nepenthales with regard to the formation and further processing of endocytotic vesicles upon stimulation of the traps with fluorescence-labeled proteins. Special attention is paid to the dynamics of nutrient uptake, and the hypothesis is tested that species with permanent or temporal digestive cavities (*Nepenthes* and *Dionaea*/*Aldrovanda*, respectively) differ from those species with adhesive traps, where the prey is localized on the surface of the trap and might therefore be lost or removed. Furthermore, an attempt is made to quantify endocytotic nutrient uptake and to assess its contribution to the overall mineral supply of Nepenthales CPs.

## 2. Results

### 2.1. Quantification of Glandular Cells

The investigated species vary in trapping mechanisms as well as trap sizes, and differ widely with regard to number of glands and glandular cells per trap. *D. roseana* possesses 107 ± 36 stalked glands and a total of 3817 ± 1319 glandular cells per trap, whereas *N.* × *ventrata* exhibits 19,139 ± 3128 sessile glands and 3,130,519 ± 635,483 glandular cells. Table 1 characterizes the five species in quantitative terms. For the estimation of protein uptake, only the arithmetic means of cells per digestive zone were used (DZ, defined in Section 4.2).

### 2.2. Endocytosis on the Cellular Level

Endocytosis was quantified in a total of 12,076 cells (1398 in sessile glands of *D. lusitanicum*, 1408 in tentacles of *D. lusitanicum*, 4199 in *D. capensis*, 3445 in *D. roseana*, 1385 in *D. muscipula*, and 241 in *N.* × *ventrata*). For each species and time point, vesicles were quantified in 10 randomly selected glandular cells. In both species of *Drosera*, endocytotic vesicles were observed after 15 min of incubation. In the stalked and sessile glands of *D. lusitanicum*, vesicles became visible after 60 min (Figure 1), whereas in *N.* × *ventrata* and *D. muscipula* vesicle formation was only noted after 16 h. Thus, vesicles appeared considerably earlier in adhesive traps compared to traps forming a temporary stomach (*D. muscipula*) or a permanent pool of digestive fluid (*N.* × *ventrata*). Throughout the observation period, the number of vesicles remained constant in *D. capensis* (R^2^ = 0.00; *p* = 0.691; 9.7 ± 7.0 vesicles per cell). In *D. muscipula*, vesicle number decreased significantly over time (R^2^ = 0.37; *p* < 0.001), from 42.9 ± 8.5 vesicles per cell at 16 h to 20.1 ± 6.3 at 64 h. In all other species, the vesicle number decreased slightly but significantly over time (R^2^ < 0.1; *p* < 0.05). Typically, fewer than 20 vesicles were found (Figure 2). No obvious difference was found between sessile and stalked glands.

Simultaneous to the decreasing vesicle number, an increase in vesicle diameter over time was observed in *D. muscipula* (R^2^ = 0.34; *p* < 0.001; from 0.5 ± 0.2 µm to 1.6 ± 1.0 µm). In *D. capensis* the vesicle diameter remained constant (R^2^ = 0.00; *p* = 0.777; 1.3 ± 0.7 µm). In the other three species, the trend towards increased diameter was consistently highly significant (*p* < 0.001) but less pronounced (R^2^ ≈ 0.20). Almost all vesicles were spherical, allowing for an estimate of vesicle volume. Therefore, the same trends were found for vesicle volume as for vesicle diameter, but upper outliers were more pronounced for vesicle volume. Appendix A shows the diameters and Appendix A shows the volumes.

Fluorescence intensity of the vesicles (brightness corrected for different gain settings) decreased slightly but significantly over time in *D. muscipula* (R^2^ = 0.10; *p* < 0.001; from 37 ± 12.7 arbitrary units/AU to 25 ± 7.8 AU). In *D. capensis*, on the other hand, a strong increase of the fluorescence intensity was observed (R^2^ = 0.48; *p* < 0.001; from 10.4 ± 6.4 AU to 66.7 ± 25.1 AU). In the other species, a similar but weaker trend towards increasing fluorescence intensity was observed (R^2^ < 0.2; *p* < 0.001) (Appendix A).

10 mg/mL FITC-BSA were used to stimulate uptake. Based on vesicle size and fluorescence intensity, FITC-BSA concentration within the vesicles was estimated. Except for a few outliers, all estimated concentrations were <10 mg/mL. In the vesicles of *D. muscipula*, the estimated FITC-BSA concentration decreased significantly over time (R^2^ < 0.31; *p* < 0.001; from 8.4 ± 0.9 mg/mL to 6.4 ± 0.7 mg/mL). In *D. capensis*, the estimated FITC-BSA concentration increased significantly over time (R^2^ < 0.48; *p* < 0.001; from 5.2 ± 0.7 mg/mL to 9.9 ± 1.2 mg/mL). The trend toward increasing concentration within vesicles was also observed in the other species (R^2^ < 0.15; *p* < 0.001), except in *N.* × *ventrata*. In this species the concentration stayed constant through time at a value of 6.2 ± 1.3 (Figure 3).

### 2.3. Quantification of Protein Uptake

In all species, an increase in FITC-BSA concentration multiplied by total vesicle volume per cell was observed. Thus, the highest amount of uptake was found at the end of the experiment (after 64 h). Table 2 shows the total vesicle volume and the amount of absorbed FITC-BSA by cell, gland, and DZ.

The analysis of temporal changes in entire glands delineated a clear upward trend of a volume increase of the FITC-BSA vesicles. This tendency was the most pronounced in *D. muscipula*’s glands (R^2^ = 0.66; *p* < 0.001); here the volume rose from 162 ± 75.9 µm^3^ to 2810 ± 1644 µm^3^. In the sessile glands of *D. lusitanicum* and *N.* × *ventrata*, a similar trend of volume increase was observed (R^2^ < 0.22; *p* < 0.001). The volumes increased from 841.8 ± 683.4 µm^3^ to 1940 ± 3485 µm^3^ in the sessile glands of *D. lusitanicum* and from 2369 ± 3133 µm^3^ to 8747 ± 7136 µm^3^ in *N.* × *ventrata*. The FITC-BSA volumes increased significantly in *D. roseana* (R^2^ = 0.11; *p* < 0.001), exhibiting volumes of 10.4 ± 86.8 µm^3^ in the beginning to 748.4 ± 776.9 µm^3^ at the end of the experiment. In the stalked glands of *D. capensis* and *D. lusitanicum* the volumes stayed consistent, with 6169 ± 8960 µm^3^ in *D. lusitanicum* and 4615 ± 4313 µm^3^ in *D. capensis*. Appendix A shows temporal changes of vesicle volumes in glands.

The volume of vesicles and the estimated concentration of FITC-BSA, allowed for an estimation of total FITC-BSA content. Overall, the absorbed amount increased inside glands throughout the observed timeframe, except in the sessile glands of *D. lusitanicum*; here the content remained constant at a level of 33.9 ± 53.6 ng. Most of the variance was explained in *D. muscipula* (R^2^ = 0.63; *p* < 0.001), where 1.4 ± 0.7 ng of FITC-BSA at the beginning of the experiment accumulated to 17.5 ± 10.7 ng at 64 h. A similarly pronounced increase was observed in the stalked glands of *D. lusitanicum* (R^2^ = 0.25; *p* < 0.001), with 0.5 ± 1 ng at the beginning of experimentation and 30.1 ± 55.1 ng at the end. In *N.* × *ventrata*, the FITC-BSA contents also rose significantly over time (R^2^ = 0.20; *p* < 0.001), from 14.5 ± 18.1 ng at the first observation after an incubation time of 2 h to 47.8 ± 44.3 ng at 64 h. The FITC-BSA content in *D. roseana* increased significantly from 0 ± 0.3 ng to 5.9 ± 6.3 ng (R^2^ = 0.17; *p* < 0.001). The initial 5.4 ± 10 ng of FITC-BSA on glands *D. capensis* increased to a content of 39.9 ± 55.2 ng throughout the duration of the experiment (R^2^ = 0.07; *p* < 0.001). Appendix A shows temporal changes of the FITC-BSA content in glands.

Estimates for vesicle volume and FITC-BSA per gland and DZ were achieved by multiplying the estimates per cell with the number of cells. Temporal changes were therefore identical with the trends shown above (Figure 4 and Appendix A).

The overall uptake of nutrients was assessed by the N (155.8 mg/g) and S (17.8 mg/g) content of FITC-BSA. Uptake rates were estimated by the slope of the linear regression models in order to compensate for different latency times. In the sessile glands of *D. lusitanicum*, 12 ng/d N and 1 ng/d S was absorbed, but these uptake rates did not significantly differ from zero. In all tentacles of adhesive traps, significant rates of uptake were found, whereby uptake rates were roughly correlated with trap size: 36 ng/d N and 4 ng/d S in *D. roseana*, 135 ng/d N and 15 ng/d S in *D. lusitanicum*, and 475 ng/d N and 54 ng/d S in *D. capensis*. In both species with temporary stomachs, absorption rates were far higher: 8.3 µg/d N and 0.9 µg/d S in *D. muscipula* and 63.6 µg/d N and 7.3 µg/d S in *N.* × *ventrata*.

The results of all regression analyses are in the Appendix A.

## 3. Discussion

### 3.1. Endocytosis in Carnivorous Plants

Carnivorous plants *sensu stricto* digest their prey, predominantly by means of proteolytic enzymes. Such enzymes are usually excreted by exocytosis; in order to keep membrane surface constant, exocytosis is always linked to endocytosis [23,24,25], which leads to incorporation of external media into the cytoplasm of glandular cells. Thus, endocytotic uptake of nutrients by carnivorous plants was already predicted in 1977 [18], experimental evidence was only found after the development of confocal laser scanning microscopy [19]. Thereby, membrane impermeable dyes (Lucifer Yellow), membrane dyes (FM 1-43), and the fluorescent-labeled protein analogues FITC-BSA, and TRITC-BSA (unpublished data) consistently resulted in the formation of vesicles, which tended to grow in size over time. Endocytosis was found in all investigated species of Nepenthales, as well as in *Cephalotus follicularis* and some species of *Sarracenia*, but not in Lentibulariaceae and other Sarraceniaceae [19,20]. However, it remained unclear if endocytosis is only a side effect of enzyme secretion or significantly contributes to nutrient uptake.

This study aimed to quantify endocytotic protein uptake, by using 1% FITC-BSA, which efficiently stimulates endocytosis in all Nepenthales and where the fluorescence intensity allows for an estimate of the amount of protein. In contrast to the previous studies [19,20], no attempt was made to optimize experimental conditions for each species. Our experimental setups were homogenized, especially regarding constant temperatures (27 °C) throughout the experiments. This explains the most obvious difference to the mentioned previous experimentations i.e., the extended latency until first vesicles were formed. Furthermore, Lichtscheidl et al. could show in a similar staining procedure that glandular heads in *D. capensis* take up fluorescent dye visible in small fluorescent vesicles situated at the plasma membrane. These rather small vesicles grow to a specific size and fuse to form multivesicular bodies [21]. In contrast to our experiments, their observations incorporated a substantially smaller sample size, analyses were made over shorter monitoring intervals, and incubation times were only 3–6 h at maximum.

### 3.2. Quantitative Assessment for Uptake via Endocytosis on a Cellular Level

Our standardized approach allowed for quantitative characterization of trends that so far had only been described in qualitative terms. Though qualitative trends were similar in all investigated species, significant quantitative differences were identified. In all adhesive traps, formation of vesicles started rapidly, in less than 15 min in both *Drosera* species and after 45 min in both the stalked and sessile glands of *D. lusitanicum*. In *Drosera* secretion of mucilage or at least water may be an ongoing process [26,27], so that FITC-BSA was taken up as soon as it was available in the medium. The trapping mucilage of *D. lusitanicum* is much less prone to exsiccation [28] and therefore may not require constant replacement causing a slightly delayed start of endocytosis (60 min). In contrast, similarly to animals [3], *D. muscipula* and *N.* × *ventrata* form temporary stomachs [29,30], where the glands are more or less inactive until prey is trapped [30], which is in good consistency with their observed endocytosis latency of 16 h.

In all species, except for *D. capensis*, the vesicle number decreased significantly over time and vesicle size increased, suggesting fusion of smaller endocytotic vesicles into larger endosomes. This trend was most pronounced in *D. muscipula*. The constant vesicle size after 15 min observed in *D. capensis* seems to be in contrast to the results of Lichtscheidl et al., where an increase of vesicle size up to formation of multivesicular bodies (already after 10 min) with a specific diameter was observed. In our CLSM observations those multivesicular bodies were the only vesicle types that were traced. Their diameter of 1.3 ± 0.7 µm remained constant over time. Both the very first tiny FITC-BSA vesicles marking the start of endocytosis and their fusion to larger vesicles such as in the experimentations of Lichtscheidl et al. could not be resolved in our study due to the lower numerical aperture of the objective and standardized microscopic setup. Moreover, discrepancies between results are caused by variances of sample size, staining treatment, incubation time, and monitoring intervals [21].

In our examinations, fluorescence intensity increased over time in all species except *D. muscipula*, and was most pronounced in *D. capensis*. Thus, on-going endocytosis seems to lead to accumulation of higher concentrations of FITC-BSA in vesicles of constant size and number in *D. capensis*. This is in accordance with the estimated concentrations of FITC-BSA inside vesicles; though fluorescence intensity depended not only on FITC-BSA concentration but also on vesicle size, probably due to the non-zero thickness of the focal plane of the CLSM and also fluorescence resonance energy transfer within each vesicle. Concentrations were constantly <10 mg/mL, i.e., smaller than the offered nutrient solution. Obviously, exocytotic vesicles are not emptied completely into the external medium and are therefore only partially filled with the FITC-BSA solution [24,25], suggesting a “kiss-and-run” type of endocytosis [23]. Over time, the FITC-BSA concentration within the vesicles increased significantly in all adhesive traps, especially in *D. capensis*. Though there seems to be a mechanism to remove water from the vesicles, concentrations exceeding the external medium were found only in exceptional cases. In traps with a temporary stomach, concentrations within vesicles decreased over time or remained constant, possibly due to fusion with primary lysosomes in order to enable digestion of the protein. The same process may be active in adhesive traps as well but would be masked by the removals of water from vesicles.

### 3.3. Quantification of Endocytosis at Gland and Trap Level

Based on the number and volume of vesicles and on the FITC-BSA concentration within, the uptake of whole glands and entire traps could be estimated. Thereby, the number of cells per gland and the number of glands per trap were determined. This was done either by counting, or by approximating the shape of glands and traps by geometrical bodies. In addition to the unavoidable inaccuracies of these estimates, this approach assumes similar uptake rates for all gland cells of a trap. Within each gland, cells showed indeed similar uptake patterns. However, the assumption that all glands within a trap contribute to uptake equally and follow the same processes is not obvious. In *N. × ventrata* all submerged glands are exposed to the same digestive fluid, but the fluid level is not necessarily constant. Furthermore cells at the bottom and the upper parts of the pitcher differ in size and shape [31]. In the linear trap leaves of *D. capensis* and *D. lusitanicum*, only a section of the trap will come into contact with an individual prey object. Here, a rather arbitrary length of 5 mm was assumed for the immotile traps of *D. lusitanicum* and 1 cm for the traps of *D. capensis*, which will fold around the prey. Only in *D. roseana* and *D. muscipula* can it reasonably be assumed that all glands will come in contact with the dissolved prey. Finally, this study focuses exclusively on the FITC-BSA present in the glandular cells. Further transport beyond gland cells was not taken into account. Therefore, the values for total uptake are to be understood as rough estimates for total trap uptake at minimum.

Throughout the experiment, the FITC-BSA content per cell increased significantly in all species, except in the sessile glands of *D. lusitanicum*. The highest uptake rate (18 pg/h) was found in *D. muscipula*. After 64 h, the amount of FITC-BSA per cell differed significantly (*p* < 0.001, Kruskal-Wallis test) between the species, ranging from 0.1 ± 0.1 ng in *D. lusitanicum* to 0.7 ± 0.4 ng in *D. muscipula*. Thus, the sessile glands of *D. lusitanicum* contribute very little to endocytotic uptake, especially compared to the tentacles (0.7 ± 1.2 ng). In this gland type, the FITC-BSA uptake might be a side effect of ephemeral secretion of digestive enzymes [32] upon stimulation, whereas the tentacles are responsible for endocytotic nutrient uptake.

The uptake per gland differed more strongly due to the differences in gland size and ranged from 5.9 ± 6.3 ng in *D. roseana* to 47.8 ± 44.3 ng in *N.* × *ventrata*. Extreme differences were found for the uptake by trap, where *D. roseana* with 3817 ± 1319 glandular cells per trap absorbed 632 ± 681 ng throughout the experiment, whereas *N.* × *ventrata* with 3,130,000 ± 635,000 glandular cells absorbed 915 ± 8.47 µg. *D. muscipula* however, absorbed 90 ± 56 µg in spite of its much smaller traps. In terms of uptake rates over time, these values equal 36 to 63,000 ng/d N. Little information is available on uptake rates per day and trap.

### 3.4. Relevant N and S Nutrient Supply via Endocytosis

In *Sarracenia purpurea*, mutualistic trap inquilines release 9100 ng soluble N per trap and day, which is assumed to be absorbed by the trap [33]. Given the size of *S. purpurea* traps exceeding traps of *D. muscipula* but being much smaller than *N.* × *ventrata* our values appear plausible in comparison.

Due to the very different sizes of the five investigated species, comparisons of uptake rates are difficult. However, the very high uptake rate and the high amount of absorbed FITC-BSA in *D. muscipula* is in good accordance with estimates of 80% dependency on animal prey for N supply in this species [34]. In addition, for *N.* × *ventrata* a high dependency on animal prey can be assumed since Schulze et al. (1997) reported 62% for *N. mirabilis* [35] Similarly, the much lower uptake rates in both *Drosera* species fit well with only 18% of prey derived N in rosette *Drosera*, reported by Schulze et al. (1991).

In addition to N, S can also be derived from the FITC-BSA and other proteins, roughly one order of magnitude below N. Unlike uptake by carrier proteins for ions or amino acids, endocytosis inevitably leads to the uptake of whole proteins including its S; accordingly, growth propagation by prey capture in sulfur-deficient plants has been demonstrated in *D. binata* [14].

The uptake of ammonium and amino acids by carrier proteins has been demonstrated for *Nepenthes* [34] and is highly probable for other Nepenthales carnivores. Remarkably no carrier proteins for peptides have been found in *Nepenthes* glands; thus, carrier proteins would absorb prey-derived nutrients only after a complete-break down of all proteins. In spite of the well-known digestive activity of the pitcher fluid and its inquilines [34], a complete break-down of proteins appears unlikely. In the other investigated species, the whole process of prey utilization takes much less time and does not seem to involve mutualistic symbionts, rendering complete digestion to amino acids even less plausible. The very significant amounts of protein absorbed by endocytosis shows that endocytotic nutrient uptake has the potential to supplement carrier proteins in a substantial manner.

## 4. Materials and Methods

### 4.1. Experimental Setup

*Drosera roseana*, *Dionaea muscipula* and *Nepenthes* × *ventrata* were cultivated in the greenhouses of the HBLFA for Horticulture and the Bundesgärten Schönbrunn. *Drosera capensis* was cultivated at the former Biocenter of the University of Vienna (Althanstraße 14) in the greenhouse of the faculty of Molecular Systems Biology (MoSys). *Drosophyllum lusitanicum* was cultivated by Stefanie Krammer.

FITC-BSA (Albumin-fluorescein isothiocyanate conjugate, Sigma-Aldrich, St. Louis, MO, USA, Prod. Nr. A 9771) is a bovine serum albumin labeled with the fluorescent dye FITC and can be used to test nutrient uptake via endocytotic vesicles in various carnivorous plants. BSA mimics prey protein, therefore it is able to trigger uptake and digestion [19,20,21,36].

The FITC-BSA powder was first dissolved in distilled H_2_O into a 5% FITC-BSA stock solution—followed by a further dilution with pitcher fluid for *Nepenthes* × *ventrata* or distilled water for the other species to a final staining solution of 1% FITC-BSA. For sample preparation for short term incubation (15 min–32 h), trap leaves were cut off and the staining solution was applied on the plant tissue and subsequently stored in a dark and humid chamber at 27 ± 1 °C. For long-term incubation (>32 h), the staining solution was directly applied in vivo on the trap leaves. The stained areas were covered with thin aluminum foil to prevent photobleaching. A more detailed preparation protocol is provided in Table 3.

All samples were analyzed using the Confocal laser scanning microscope (CLSM) Leica TCS SP5 DM-6000 CS. FITC-BSA fluorescence was excited at 488 nm laser excitation and detected between 505–540 nm. The different incubation times are shown in Table 4.

In the CLSM, size and fluorescence intensity of endocytotic vesicles in glandular cells were measured:Vesicle diameterVesicle brightness on 0–256 greyscale valuesNumber of vesicles per cell


Transport of fluorescent dye into stalk and tissue was not considered.

### 4.2. Calculation of Cells per Trap Leaf/Digegestive Zone

In order to approximate total uptake by glandular heads in whole trap leaves, the total number of gland cells per gland and in extension trap leaf (digestive zone) was calculated. The estimation of nutrient uptake was determined for entire traps in *D. muscipula* and *N.* × *ventrata*, as these two species digest their prey within a digestive chamber. Here, every gland can be expected to be in contact with prey since the digestive fluid covers the whole digestion zone.

To calculate the surface area of the digestive zone in *D. muscipula*, each lobe of the trap leaf was assumed to have an elliptic form. Therefore, the formula used was:(1)$$A=\frac{\pi \times l\times h}{2}$$
where *l* was the longest diameter of the lobe and *h* the shortest. A total of 15 leaves were measured to get an estimate. For calculation of the digestive zone in *N.* × *ventrata*, the form of half of a prolate spheroid was assumed and its area defined in 5 pitchers by the following formula:(2)$$A=\pi \times d\times \left[d+\left(\frac{h^{2}}{\sqrt{h^{2}-d^{2}}}\right)arcsin\left(\frac{\sqrt{h^{2}-d^{2}}}{h}\right)\right]\;$$
where *h* was the height of the digestive zone as far as covered by pitcher fluid, and *d*, radius of the pitcher.

Subsequently the number of glands per cm^2^ was evaluated in 3 traps of *D. muscipula* and *N.* × *ventrata* respectively. This gland density was then multiplied by the area of the digestive zones to deliver the total number of glands that can come in contact with prey.

In *Drosera* and *Drosophyllum*, only glands close to the prey secrete digestive fluid and participate in nutrient uptake. Thus, we predefined the area in which digestion and uptake are performed: In *Drosera* the stalked glands, tentacles, and the entire leaf lamina are able to bend toward and around prey. Thus, considering the small size of trap leaves in *D. roseana*, we assumed that all tentacles ultimately come in contact with prey within. The total number of tentacles in 11 trap leaves was counted. For *D. capensis* we assumed that 1 cm leaf length corresponded to the digestive zone, which would perform uptake from a single prey object. Here, the mean number of tentacles per cm was determined in 3 leaves. The stalked glands and trap leaves of *D. lusitanicum* are not motile, so a digestive zone comprising a leaf length of 0.5 cm was assumed. Here, the number of stalked and sessile glands in 11 different positions of two trap leaves was counted using a Wild Photomacroscope M400.

Electron microscopy was performed for 3D geometrical modeling of the glandular heads. Samples were prepared for scanning electron microscopy (SEM: JEOL IT 300) and for field emission microscopy (FESEM: JEOL JSM 6330F Field Emission Scanning Microscope).

For SEM, fresh plant leaves were sectioned, transferred in PBS (Phosphate-buffered saline) and air from plant tissue was gently evacuated. Fixation was performed with 4% formaldehyde in 10% PBS for 6 h, and during the preparation steps motion of liquids in test tubes was maintained by a lab shaker. Fixed samples were then rinsed three times in 10% PBS and multiple times in ddH_2_O. Subsequently, the samples were soaked in ddH_2_O for 17 h. Plant material was then transferred into acidified Dimethoxy-propane for 30 min. Before critical point drying (Leica EM CPD300), samples were thoroughly washed with acetone. Dried plant material was mounted on stubs and sputter coated with gold.

For FESEM, fresh plant material was sectioned and placed in carbon paste on a stub, followed by quick freeze fixation in a mixture of liquid and solid nitrogen (slush). Then, the fixed samples were transferred into a preparation chamber (Oxford-Alto2500 cryo-system), freeze-etched to remove water from the surface, and sputter coated with gold.

The next step in calculation of number of cells per DZ, comprised counting and measuring of glandular cells in the light microscope. The number of cells per gland was determined in 7 digestive glands in *N.* × *ventrata* and 10 of *D. muscipula*. Due to the curvature of the gland surface, the number of cells within sessile and stalked glands in *D. roseana*, *D. capensis*, and *D. lusitanicum* was established by calculation of total gland area divided by cell area (*n* = 10 for gland and cell area in *Drosera*). In stalked glands of *D. lusitanicum* the shape of the glandular head was approximated by half of an oblate spheroid, whereby the calculation of the gland area was performed for 8 glands.

The shape of sessile glands was assumed as an irregular ellipsoid where the surface area (of *n* = 11) was approximated after Thomson [37], whereby the area of the joint was subtracted.

Tentacle head surface areas in *D. roseana* and *D. capensis* where calculated through addition of the area of one half of a prolate spheroid and a truncated cone. Figure 5 describes how morphological appearance was translated in 3D geometrical model and subsequently into surface area formulas.

Standard deviations (*SD*) for the number of cells per gland and per trap were not calculated directly but derived from the confidence intervals (at appropriate levels, with *UL_CI_* − *LL_CI_* being the width of the confidence interval) of the surface areas of cells and glands, following the formula:(3)$$SD=\frac{(UL_{CI}-LL_{CI})\times \sqrt{n}}{2z}$$

### 4.3. Quantification of FITC-BSA Uptake

Based on the fluorescence intensity, the amount of protein absorbed by endocytosis was estimated.

Due to the significant differences in fluorescence intesity some of the images were acquired using an approximate photomultiplier voltage of 500 V or 700 V. In order to standardize the fluorescence intensities, 53 vesicles were measured at both photomultiplier settings. To produce vesicles in various sizes, a FITC-BSA solution was suspended in immersion oil. Standardized brightness for an assumed photomultiplier voltage of 500 V was established through the following formula:(4)$$B_{stand}\;=\;B\times (\mathrm{Gain}\times (-0.0036)+2.80)$$

Possible deviations from a linear relationship between gain and brightness were considered negligeable because only gains very close to 500 V or 700 V were used.

In order to assess the FITC-BSA concentration inside a vesicle, FITC-BSA solutions ranging from 0.1% to 1.2% were suspended in immersion oil and their brightness and diameter were measured. It was found that vesicle brightness depended in a complex manner both on FITC-BSA concentration and on vesicle size. The best model explained 79% (R^2^) of the concentration and was highly significant (Equation (4)).
(5)$$Conc_{FITC-BSA}=0.30+0.11\times \sqrt{B_{stand}}+\left(-0.15\right)\times \ln\left(diameter\right)$$

Stata’s predict command was used to calculate FITC-BSA concentration for vesicles mesured in vivo. FITC-BSA content of vesicles was calculated by concentration multiplied by volume; FITC-BSA content of cells was calculated by summing up all vesicles in an individual cell.

The summary formula of *BSA* is C_2932_H_4614_N_780_O_898_S_39_ [38], resulting in a molecular weight of 66,430.3 g/mol [39]. The FITC conjugate of BSA contains 7–12 moles FITC (molecular weight: 389.4) per mole BSA, resulting in a total molecular weight of 70,129.4. It was assumed that only BSA derived N and S were accessible for the plants, resulting in 155.8 mg N and 17.8 mg S per gram FITC-BSA [40].

### 4.4. Statistics

MS Excel was used for data management and for the calculation of cell numbers per gland and trap. For all other analyses, Stata 14 was used.

All data were summarized by descriptive statistics (arthmetic mean, SD, 95% confidence interval, 1st to 3rd quartile, and sample size), itemized by timepoint and subgroups. Subgroups were formed by species (in *D. lusitanicum*, sessile and stalked glands were treated seperately), by trap type (adhesive traps vs. temporary stomachs and adhesive traps, snap-traps, and pitcher traps), and by gland type (sessile vs stalked glands). Differences between subgroups were tested for significance by the non-parametric Kruskal-Wallis test.

Trends over time were investigated by linear regression models for each subgroup. Here, the significance and R^2^ for each model are reported, together with coefficients (including confidence intervals) and *p*-values for time and the constant term.

## 5. Conclusions

Endocytotic uptake of FITC-BSA starts almost immediately in adhesive traps but with a significant delay of up to 16 h in *D. muscipula* and *N.* × *ventrata*.Over time, FITC-BSA accumulates in glandular cells, either by fusion of small endocytotic vesicles to larger compartments, or, in *D. capensis* by increasing FITC-BSA concentration within the vesicles. However, the vesicular concentration of FITC-BSA does not exceed that in the external medium.After 64 h, estimated uptake per trap ranges from 263 ng in *D. roseana* to 915 µg in *N.* × *ventrata*, whereby most of the difference is explained by trap size.In *D. lusitanicum* sessile glands contribute very little to endocytotic nutrient uptake compared to the tentacles.Estimated uptake rates per trap range from 36 ng/d N in *D. roseana* to 63 µg/d in *N.* × *ventrata*. Differences between species are in accordance with the available information on prey dependency for nutrient supply.Endocytotic nutrient uptake seems to play an equally important role as uptake via carrier proteins in the observed species. Furthermore, endocytotic nutrient uptake requires a far less efficient digestion since whole protein molecules can be absorbed.

## Figures and Tables

**Figure 1 plants-12-00341-f001:**
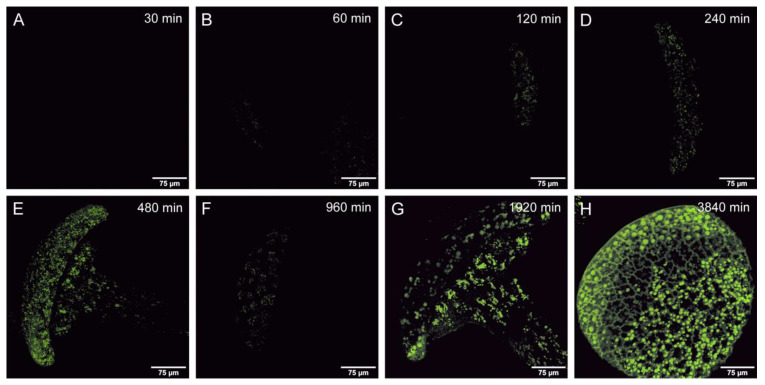
CLSM images of stalked glands of *Drosophyllum lusitanicum*; 1% FITC BSA treated, after an incubation time of: (**A**): 30 min, (**B**): 1 h, (**C**): 2 h, (**D**): 4 h, (**E**): 8 h, (**F**): 16 h, (**G**): 32 h, (**H**): 64 h.

**Figure 2 plants-12-00341-f002:**
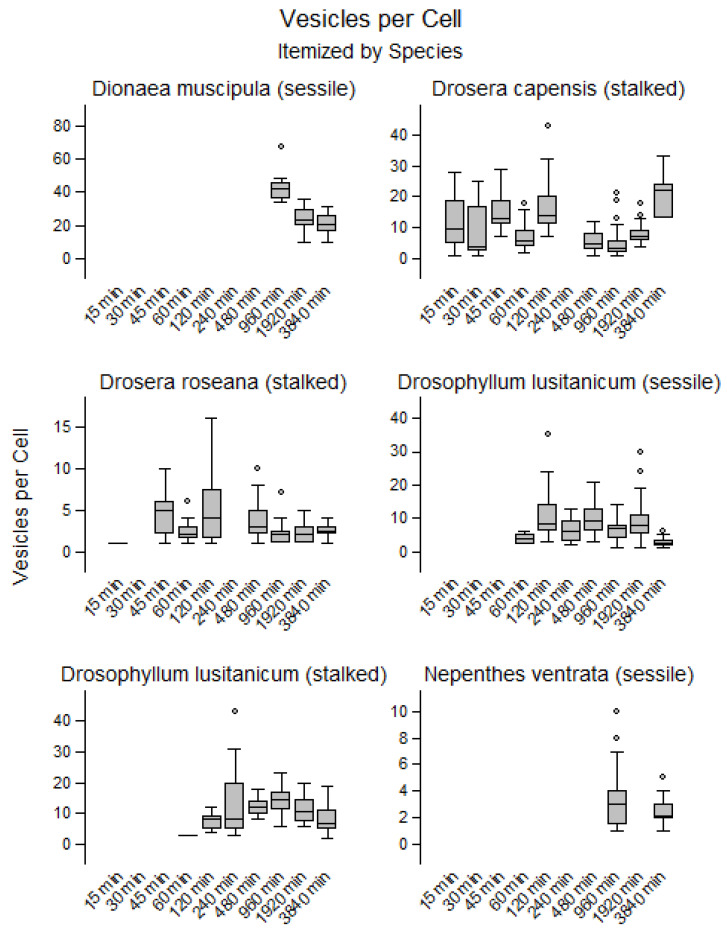
Temporal changes of vesicle numbers in observed glandular cells in *Dionaea muscipula*, *Drosera roseana*, *Drosera capensis*, *Nepenthes* × *ventrata* and stalked and sessile glands in *Drosophyllum lusitanicum*. In all other species, vesicle number decreased slightly but significantly over time (R^2^ < 0.1; *p* < 0.05), only in *D. muscipula*, vesicle number decreased substantially over time.

**Figure 3 plants-12-00341-f003:**
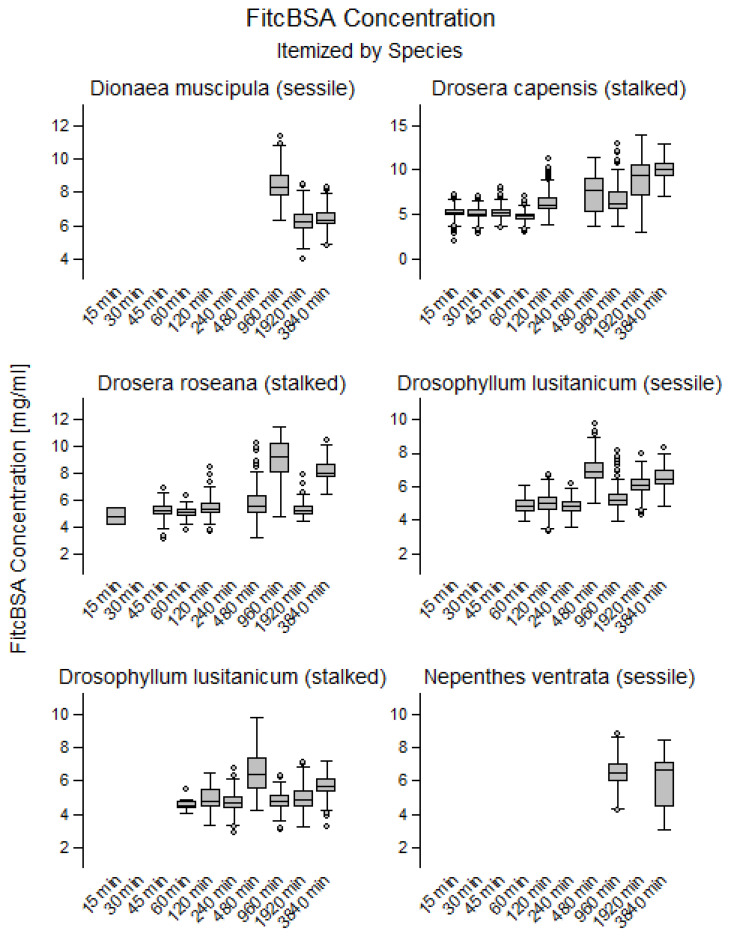
Temporal changes of FITC-BSA concentrations in observed glands in *Dionaea muscipula*, *Drosera roseana*, *Drosera capensis*, *Nepenthes* × *ventrata* and stalked and sessile glands in *Drosophyllum lusitanicum*. Estimated FITC-BSA concentrations within vesicles exhibited a trend toward increase in all species with the exception of *N.* × *ventrata* and *D. muscipula*. The concentrations in *N.* × *ventrata* stayed constant, whereas in *D. muscipula* concentration decreased significantly over time.

**Figure 4 plants-12-00341-f004:**
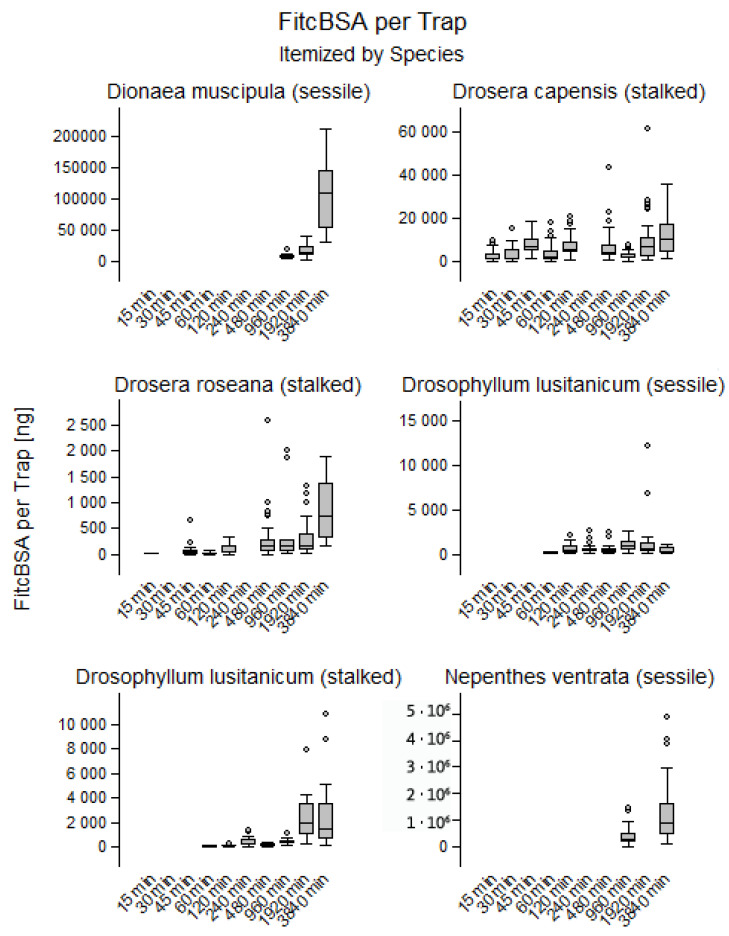
Temporal changes of total FITC-BSA content per trap (DZ) in *Dionaea muscipula*, *Drosera roseana*, *Drosera capensis*, *Nepenthes* × *ventrata* and stalked and sessile glands in *Drosophyllum lusitanicum*. Throughout the observed timeframe, the amounts of FITC-BSA increased in all species, except in sessile glands of *D. lusitanicum*; here the content remained constant at a level.

**Figure 5 plants-12-00341-f005:**
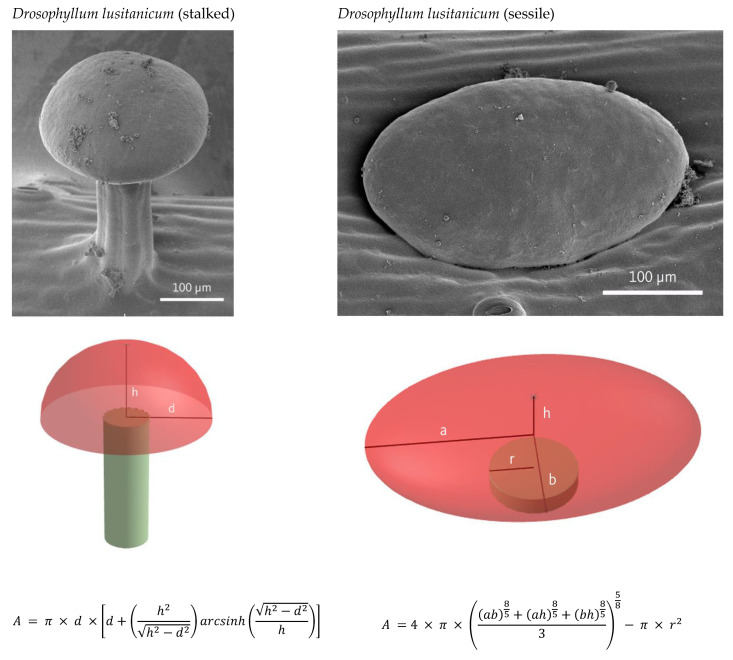
Morphology of the glands illustrated by SEM (*Drosera capensis*, *Drosera roseana)* and FESEM (*Drosophyllum lusitanicum*), their approximation by geometric bodies, and formulas used for calculation of glandular surface areas (visible in shades of red).

**Table 1 plants-12-00341-t001:** Quantification of digestive glands and glandular cells (arithmetic mean ± standard deviation). DZ: digestive zone, comprising entire traps in *D. roseana*, *D. muscipula*, *N.* × *ventrata*, and 1 cm in *D. capensis*, 0.5 cm in *D. lusitanicum*. Gland area and cell area of *D. muscipula* and *N.* × *ventrata* are not filled out since number of cells per gland was counted directly.

	*Drosophyllum lusitanicum*		*Drosera capensis*	*Drosera roseana*	*Dionaea muscipula*	*Nepenthes* × *ventrata*
Gland types	sessile	stalked	stalked	stalked	sessile	sessile
Gland Area [µm^2^]	26,294 ± 14,708	94,534 ± 57,135	46,576 ± 11,955	10,232 ± 9108		
Cell Area [µm^2^]	570 ± 106	545 ± 100	223 ± 61	288 ± 71		
Cells per Gland	46 ± 65	174 ± 9	208 ± 47	36 ± 17	24 ± 6	164 ± 53
Area DZ [mm^2^]	12 ± 2		27 ± 2		427 ± 70	4161 ± 831
Glands/DZ ^1^	41 ± 5	21 ± 2	225 ± 19	107 ± 36	5213 ± 1101	19,139 ± 3128
Cells/DZ ^1^	1891 ± 1149	3711 ± 265	46,864 ± 6103	3817 ± 1319	126,675 ± 25,277	3,130,519 ± 635,483

**Table 2 plants-12-00341-t002:** Estimated FITC-BSA uptake after 64 h of incubation by vesicle, cell, gland, and DZ, itemized by species.

	*Drosophyllum lusitanicum*		*Drosera capensis* ^2^	*Drosera roseana* ^2^	*Dionaea muscipula* ^1^	*Nepenthes* × *ventrata* ^1^
Gland types	sessile	stalked	stalked	stalked	sessile	sessile
Sample size	56	242	209	24	420	121
Volume per Cell [µm^3^]	11.2 ± 20.1	119.6 ± 240.1	18.6 ± 24.5	21.1 ± 21.9	115 ± 67.3	53.5 ± 43.6
FITC-BSA per Cell [ng]	0.1 ± 0.1	0.7 ± 1.2	0.2 ± 0.3	0.2 ± 0.2	0.7 ± 0.4	0.3 ± 0.3
Volume per Gland [µm^3^]	1940± 3485	5517 ± 11075	3879 ± 5099	748 ± 777	2810 ± 1644	8747 ± 7135
FITC-BSA per Gland [ng]	12.3 ± 23.1	30.1 ± 55.1	39.9 ± 55.2	5.9 ± 6.3	17.5 ± 10.7	47.8 ± 44.3
Volume per DZ [µm^3^]	41,489 ± 74,525	226,159 ± 454,008	871,926 ± 1146,124	80,352 ± 83,406	14,648,188 ± 8,570,270	167,406,304 ± 136,562,528
FITC-BSA per DZ [ng]	263 ± 495	1235 ± 2257	8969± 12,396	632 ± 682	91,260 ± 55,964	915,266 ± 847,190
Nitrogen ^1^ per DZ [ng/day]	12 (−21–44)	135 (94–176)	475 (295–654)	36 (24–48)	8338 (6468–10,170)	63,561 (35,893–91,603)
Sulfur ^2^ per DZ [ng/day]	1 (−2–5)	15 (11–20)	54 (34–75)	4 (3–6)	954 (740–1164)	7274 (4108–10,483)

^1^ derived from the slope of the regression model, CI (confidence intervals) in brackets. 155.8 mg protein derived N per g FITC-BSA was assumed; ^2^ 17.8 mg protein derived S per g FITC-BSA was assumed (Section 4.3).

**Table 3 plants-12-00341-t003:** Staining and preparation for microscopy itemized by trap type and plant species.

Trap Type	15 min–32 h	64 h
Adhesive traps (*Drosera capensis*, *Drosera roseana* and *Drosophyllum lusitanicum*)	Staining	
	separate the trap from the plant cover the trap with fluorescent dye transfer to a dark and humid chamber held at a constant temperature of 27 ± 1 °C by a water bath	traps remain on the plant add staining solution cover stained leaves with aluminum foil
	Preparation for microscopy	
	wash traps with tap water perform marginal and upper leaf surface sections for *D. capensis* and *D. lusitanicum* transfer whole trap of *D. roseana* and sections of *D. capensis* and *D. lusitanicum* onto glass slide	
Pitcher traps (*Nepenthes* × *ventrata*)	Staining	
	separate the trap from the plant filter digestive fluid with filter paper dilute FITC-BSA stock with digestive fluid prepare cuttings of the digestive zone as bowl shaped pieces apply fluorescent dye to each piece transfer to dark and humid chamber held at a constant temperature of 27 ± 1 °C by a water bath	trap remains on plant filter digestive fluid with filter paper dilute FITC-BSA stock with digestive fluid fill staining solution into intact pitcher cover whole pitcher with aluminum foil
	Preparation for microscopy	
	carefully rinse with tap water section leaf longitudinally and transfer onto a glass slide	
Snap traps (*Dionaea muscipula*)	Staining	
	feed un-triggered opened trap leaves with staining solution induce trap closure by mechanical stimulation of the trigger hairs separate trap from plant transfer to dark and humid chamber held at a constant temperature of 27 ± 1 °C by a water bath	feed un-triggered opened trap leaves with staining solution induce trap closure by mechanical stimulation of the trigger hairs cover traps with aluminum foil directly in vivo on the plant
	Preparation for microscopy	
	separate trap lobes carefully rinse with tap water section leaf longitudinally and transfer onto a glass slide	

**Table 4 plants-12-00341-t004:** Plant species and observed time points of the endocytosis marker treatment with FITC-BSA. Preliminary experiments had shown that no uptake occurred in *N.* × *ventrata* and *D. muscipula* during the first two hours.

Species	15 min	30 min	45 min	1 h	2 h	4 h	8 h	16 h	32 h	64 h
*Drosera capensis*	X	X	X	X	X	X	X	X	X	X
*Drosera roseana*	X	X	X	X	X	X	X	X	X	X
*Drosophyllum lusitanicum*	X	X	X	X	X	X	X	X	X	X
*Nepenthes* × *ventrata*					X	X	X	X	X	X
*Dionaea muscipula*					X	X	X	X	X	X

## Data Availability

Raw data will be made available upon request.

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
