# Peer review of "Quantification of Protein Uptake by Endocytosis in Carnivorous Nepenthales"

_plants, 2023, doi:10.3390/plants12020341_

Round 1
Reviewer 1 Report
The present study tightly follows up on previous studies on endocytotic uptake of substances and nutrients in traps of carnivorous plants conducted by the experienced authors of the Vienna University team (2012, 2019, 2021). The essence is that a substantial part of mineral nutrients from digested prey (perhaps both N, P, K, ......) may be taken up in traps of various carnivorous taxa directly by endocytosis, without being transported by membranous transporters. Here, the authors verified endocytotic nutrient uptake in 5 species of 4 genera of the order Nepenthales. The study describes the kinetics and quantity of endocytotic uptake of labelled albumin but also discriminates trap types according to the uptake kinetics. Generally, this experimental study is original, beneficial and should be published. I suppose that the paper is well written in a well-arranged form. Although the kinetics of endocytosis of labelled albumin in glands and traps of different plant species was rather variable, the study has allowed to elucidate more general features associated with protein digestion in traps in general. I have found only minor mistakes and mainy typos. My minor comments are listed below (page/line):
p.1, l.16: the abbreviation (FITC-BSA) should be added here: „.......albumin (FITC-BSA),“......
p.1, l.22: „adsorption“?? correctly: absorption or uptake.
p.2, l.79-80: The sentence is not correct, maybe a verb has been deleted.
Table 1: In the headline, perhaps n could be stated. The use of the numbers/superscripts „1“ and „2“ is a bit confusing. First, they both should be deleted from the upper raw of species names. Second, for „2“ in the body of the table, in the explanatory note, „not required“ seems to be senseless here, it might be better: „not shown“. Check both superscripts 1 and 2 to be used correctly and logically.
p.3, l.102: The name in italics.
p.4, l.108, 109: R2: as superscript.
p.4, l.111: „globe shaped“? Perhaps better: „spherical“.
p.4, l.112, p.7, l.170: „therefore“.
Table 2: Some values for N and S per DZ are negative, is it correct? What is „CI“? This abbreviation is not included in the text but the legend should be clear, thus spell out.
p.11, l.290-311: The discussed seasonal N gain coming from prey animals in carnivorous plants in cited studies mainly depends on the seasonal capture of prey. Yet I would like to ask a question whether it would be possible at least to guess a proportion of N (or S) coming from carnivory in different species, which was absorbed in traps by the described endocytic mechanism? This guess might be valuable for future studies or at least assumptions.
p.14, l.403: subtracted.
p.16, l.419: intensity.
p.16, l.433: multiplied.
p.16, l.442: management.
p.17, l.459-460: ”However, the concentration of FITC-BSA in the external medium is not exceeded.“: the sentence is not clear and must be rephrased: e.g.: ”However, the vesicular concentration of FITC-BSA does not exceed that in the external medium.“
p.18, l.545: journal name??
In conclusion, this high-quality paper merits being published after a minor revision.
Author Response
Dear Reviewer 1,
Thank you very much for your valuable comments and beneficial suggestions. The original manuscript was changed accordingly and now sounds a lot clearer and more precise. Suggested changes and answers to comments are in blue colour, additionally, the alterations to the original manuscript were added in smaller print (9 instead of 10 like the rest of the document).
Thank you for your thoughts and helpful suggestions!
Kind regards from the author team

Reviewer 2 Report
This study is somehow following up former studies of the same group where they demonstrated that besides certain carrier systems a huge part of nutrient uptake in carnivorous plants belonging to the Nepenthales is due to endocytosis. Here, the authors investigate the endocytotic uptake of FITC-labelled BSA using confocal laser scanning microscopy. They were able not only to show the qualitative uptake but also did a quantification of the uptake over time, per trap, and they showed the differences between the various trap types (pitcher traps, snap traps, adhesive traps) in the different species. The authors provide a huge and compelling body of data with adequate statistical analyses.
Although the results are not surprising and the proof of principle for nutrient uptake in carnivorous plants was already shown, the studies provides new and detailed information about quantitative aspects.
The authors have done very careful experimentation, the data are well presented and the conclusions justified.
There are only few points I would like to address:
Abstract, lines 19 and 20: replace “expect” by “except”.
Results, lines 79/80: Rephrase and make a full sentence.
Legend Fig 2 legend, line 104: I guess you mean “vesicle number” not “vesicle volume”.
Figures: Indicate significant differences in the Figures, not only in the text.
Discussion, lines 294 and 296: Schulze et al. 1997 and 2001 are missing in the Ref list.
Line 303: Ref [33] is about Dionaea, not Nepenthes; I think you mean Ref [13]; please correct.
Please cross check again all references for correctness.
Write all the time “N. x ventrata”, not “N. ventrata” because it is a known hybrid.
Author Response
Dear Reviewer 2,
Thank you very much for your valuable comments and beneficial suggestions. The original manuscript was changed accordingly and now sounds a lot clearer and more precise. Suggested changes and answers to comments are in blue colour, additionally, the alterations to the original manuscript were added in smaller print (9 instead of 10 like the rest of the document).
Thank you for your thoughts and helpful suggestions!
Kind regards from the author team
